# Assessment of the psychometric properties of self-management measurement instruments for individuals with type 2 diabetes: A systematic review protocol

José Alexandre Barbosa de Almeida[1,2], Karolinne Souza Monteiro[1,3],
Thayla Amorim Santino[2], Rêncio Bento Florêncio[4],
Ana Tereza do Nascimento Sales Figueiredo Fernandes[2], Lucien Peroni Gualdi[1,3]*

1 Post-Graduate Program in Physiotherapy, Federal University of Rio Grande do Norte, Santa Cruz, Rio Grande do Norte, Brazil, 2 Department of Physiotherapy, State University of Paraíba, Campina Grande, Paraíba, Brazil, 3 Physiotherapy of Faculty of Health Sciences of Trairi, Federal University of Rio Grande do Norte, Santa Cruz, Rio Grande do Norte, Brazil, 4 Faculty of Health Sciences of Trairi, Federal University of Rio Grande do Norte, Santa Cruz, Rio Grande do Norte, Brazil

* lucien.gualdi@ufrn.br

## Abstract

### Context and objectives

The type 2 diabetes mellitus (T2DM) negatively impacts patients' quality of life, affecting their physical and mental functioning as well as social relationships. Self-management is essential for T2DM control, as it involves self-care behaviors and self-efficacy, leading to better health outcomes such as better glycemic control. There are a variety of instruments in the literature capable of measuring self-management in T2DM population. Therefore, the aim of this review is to identify the available T2DM self-management instruments and evaluate their measurement properties, as well as to analyze their contents based on the international classification of functioning, disability and health.

### Methods

The systematic review will follow the Consensus-based Standards for the Selection of Health Measurement Instruments (COSMIN) guidelines, and its protocol has been registered in the International Prospective Register of Systematic Reviews (PROSPERO) (registration CRD42024605840). Searches will be conducted in MEDLINE, Web of Science, Scopus, PsycINFO, Embase, and CINAHL. Additionally, a manual search will be conducted in the databases: PROQOLID, PROMIS, and Medical Outcome Trust. Studies on the development and validation of patient-reported outcome measures assessing self-management in individuals with T2DM will be included, with no restrictions on language or publication date. Data extraction will use tools

**Data availability statement:** All relevant data will be available within the manuscript and its Supporting information files when the study is completed and published.

**Funding:** This study was partly financed by the Coordenação de Aperfeiçoamento de Pessoal de Nível Superior – Brasil (CAPES). Finance Code 001. This funding source had no role in the preparation or decision of the manuscript. There was no additional external funding received for this study.

**Competing interests:** The authors have declared that no competing interests exist.

recommended by COSMIN. The modified Grading of Recommendations, Assessment, Development, and Evaluation (GRADE) approach will determine the quality of the evidence. Instruments will be categorized according to COSMIN recommendations. All steps will be conducted by two independent reviewers, with a third reviewer consulted in case of discrepancies. Additionally, the content of the instruments will be analyzed and linked to the ICF.

## Discussion

This systematic review may guide researchers and healthcare professionals to choose the most suitable instrument for their target population.

## Ethics and dissemination

Ethical approval is not required, as this study is a review of published data. The results will be disseminated through publication in peer-reviewed journals.

## Introduction

According to the International Diabetes Federation (IDF), approximately 537 million adults were diagnosed with diabetes mellitus (DM) in 2021. This number is projected to reach 643 million by 2030 and 783 million by 2045, with about 90% of these cases corresponding to Type 2 Diabetes Mellitus (T2DM) [1,2]. These statistics are concerning given the chronic nature of T2DM and its negative impact on physical functioning, mental health, and social relationships. Thus, further studies should aim to reduce these impacts, as well as develop more effective assessment methods for this population [3–5].

Self-management is crucial for individuals with T2DM, as behavioral management is key to achieving goals, providing support, gaining knowledge about the disease, making informed decisions, and mastering essential skills, which positively impact their health and well-being [6–8]. Proper self-management enables individuals with T2DM to maintain good glycemic control and reduce the complications of the disease [7].

Self-management encompasses a range of self-care behaviors and self-efficacy, reflecting the ability to perform specific tasks effectively [9]. These tasks include physical activity, exercise, diet, glycemic control, daily living activities, medication adherence, and foot care [9,10]. Therefore, it is important to measure self-management levels, particularly using patient-reported outcome measures [11].

There is a variety of validated instruments available for individuals with T2DM [12] which can make choosing a suitable tool difficult. Although there are systematic reviews assessing their quality [12,13], these studies have limitations. One study [12] specifically evaluates self-management measurement instruments for this population it showed some limitations. First, it did not include instruments for general chronic diseases which may have excluded T2DM participants as well as it only included

instruments in English and Chinese languages. The second limited the inclusion of generic studies [13]. Thus, this study aims to systematically assess the measurement properties of self-management instruments for individuals with T2DM (including generic instruments used for this population), as well as to analyze the extracted content and link them to the International Classification of Functioning, Disability, and Health (ICF).

## Methods

### Study design and registration

This protocol follows the recommendations of the Consensus-based Standards for the Selection of Health Measurement Instruments (COSMIN) [14] and the Preferred Reporting Items for Systematic Review and Meta-Analysis Protocols (PRISMA-P) [15]. The PRISMA-COSMIN [16] will be used to report the full study. The protocol has been submitted to the International Prospective Register of Systematic Reviews (PROSPERO) under registration number CRD42024605840. Any changes to the systematic review will be documented in PROSPERO and published in the final study report.

### Eligibility criteria

Studies on the development and validation of self-management measurement instruments (including self-care and self-efficacy) for adults with T2DM will be included. Studies on translation and cross-cultural adaptation assessing the same construct will also be included. Additionally, studies on self-management instruments for chronic diseases will be considered, if they include adult participants with T2DM. There will be no restrictions on language or publication date. Language experts will be consulted if necessary.

   Studies with different methodological designs, such as cohort studies, clinical trials, cross-sectional studies, post-intervention analyses, and case-control studies, will be excluded. Instruments that assess more than one construct, that have self-management only as a subscale and those used for validating another instrument will also be excluded.

### Search strategy

The search will be conducted in the following databases: MEDLINE (Ovid), Web of Science, Scopus, PsycINFO (APA), Embase and CINAHL (EBSCOhost). Additionally, a manual search will be performed in questionnaire databases: PRO-QOLID (http://www.proqolid.org), PROMIS (http://www.nihpromis.org), and Medical Outcome Trust (http://www.outcomes-trust.org).

   A manual search will also be conducted in the reference lists of all primary studies, systematic reviews, and gray literature to include potential studies. These searches will be performed from the inception date up to the finalization of the systematic review, ensuring that any studies published after the initial search are identified.

   To develop search strategies, international recommendations for systematic reviews will be followed [14,17]. The following aspects will be considered: (1) construct of interest (self-management, self-care, self-efficacy); (2) target population (adults with T2DM); (3) types of instruments (self-report scales and questionnaires); and (4) measurement properties, using a combination of sensitive filters recommended by COSMIN [17]. Search strategies are detailed in S1 File.

### Screening and study selection

The search results will be imported into the reference management tool Mendeley (https://www.mendeley.com) [18], and duplicates will be removed before screening. Then, the reference list will be exported to the systematic review platform Rayyan Qatar Computing Research Institute (http://rayyan.qcri.org) [19]. The detailed selection process will be documented and presented in a PRISMA flow diagram [16,20].

   Two independent reviewers (JABA and KSM) will screen studies by title and abstract, followed by full-text review of potentially eligible studies. Ineligible studies will be identified, and reasons for exclusion will be recorded and displayed

in the PRISMA flow diagram [16,20]. In cases of disagreement, a third reviewer (LPG) will be consulted for discussion. If missing data or any relevant information is necessary, study authors will be contacted via email.

### Data extraction

A standardized and pre-piloted form will be developed for data extraction by the authors. The extracted information will include: 1) Study characteristics (title, author, year, study type, administration mode, data collection period, data collection method, inclusion and exclusion criteria, instruments used, response rate); 2) General characteristics of the target population (number of participants, age, gender, country, language, participant conditions, collection setting, education level, socioeconomic status); 3) Instrument characteristics (construct, subscales, items, version for different populations, original version for cross-cultural adaptations, scoring); 4) Measurement properties results (reliability, validity, responsiveness); 5) Evidence on questionnaire interpretability and feasibility of included instruments.

Data extraction will be performed by two independent reviewers (JABA, TAS), and in case of discrepancies, a third reviewer (ATNSFF) will be consulted to reach a consensus. If missing data or additional information is required, study authors will be contacted via email.

### Methodological quality and measurement properties assessment

Two independent reviewers (JABA, RBF) will assess the methodological quality of the studies. In cases of disagreement, a third reviewer (LPG) will be consulted. The COSMIN Risk of Bias checklist, which includes 116 items, will be used to critically appraise the methodological quality of included studies [21,22]. This tool evaluates 10 measurement properties. Each item is rated on four levels: inadequate (I), doubtful (D), adequate (A), and very good (V). The lowest rating in a section will determine the overall rating for that property [14].

Each instrument will then be classified using the updated three-point scale for good measurement properties: sufficient (+), insufficient (-), or indeterminate (?) [14,21,22].

### Content analysis of instrument items and linkage to the ICF

The content of the included instruments will be extracted, analyzed, and linked to the International Classification of Functioning, Disability, and Health (ICF). Two independent reviewers (JABA, ATNSFF) will extract the content and subsequently link it to the ICF according to standardized linking rules [23–26]. In case of discrepancies between reviewers, a third reviewer (LPG) will be consulted.

### Data synthesis

A structured narrative synthesis of the included studies' results will be developed. If a validation study of the same instrument for different populations exists, its measurement properties results will be considered as a single instrument, but specific characteristics of each version will be discussed. This combination of results will provide the overall evidence for the instrument.

Additionally, the narrative synthesis will be conducted according to the following steps [27]: 1) grouping studies by construct and measurement properties assessed; 2) describing the methodological quality of each study based on the COSMIN Risk of Bias checklist; 3) summarizing the results for each measurement property, including consistency across studies; and 4) drawing overall conclusions for each measurement property per instrument, in accordance to the COSMIN and GRADE recommendations.

Thus, two reviewers (JABA, LPG) will assess and summarize the results in groups according to the criteria for good measurement properties. The 10 measurement properties will be rated as sufficient (+), insufficient (-), inconsistent (±), or indeterminate (?). The final rating for a measurement property will correspond to its lowest rating [14,22]. A third reviewer (ATSNFF) will be consulted if necessary.

Instruments will also be grouped into the following categories: 1) General disease management; 2) Management of symptoms and complications; 3) Psychosocial impact management; 4) Lifestyle management (nutrition, smoking, physical activity, and exercise); 5) Treatment and medication management; 6) Empowerment tools (self-efficacy, self-care); 7) Disease knowledge; 8) Well-being and quality of life management; 9) Health beliefs and perceptions; 10) Obstacles and problem-solving management. This categorization will help readers identify instruments according to their relevant domains.

## Certainty of evidence and instrument recommendation

To determine the certainty of evidence, two reviewers (JABA, ATSNFF) will apply a modified approach of the Grading of Recommendations, Assessment, Development, and Evaluation (GRADE) recommended by COSMIN [14]. The classification will be based on result inconsistency, sample imprecision, and indirectness, which refers to the applicability of results to the population. Instruments will be categorized into four levels of evidence quality: high, moderate, low, or very low. If disagreement arises between the two reviewers, a third reviewer (LPG) will conduct a review.

For instrument recommendations, the COSMIN guideline will be considered [14,16]. Each instrument will be classified and justified into one of three categories: (A) The instrument is recommended for use, and the results are reliable; (B) The instrument may be recommended but requires further research to assess its quality; (C) The instrument should not be recommended. Additionally, suggestions will be provided based on feasibility considerations.

## Ethics and dissemination

This study does not require ethical committee approval, as it is based on published data. The results of this systematic review will be disseminated through publication in peer-reviewed journals and presentations at scientific conferences.

## Discussion

Study protocols on systematic reviews help prevent arbitrary decisions during study eligibility and data extraction, reducing bias risks and ensuring reliable results [28,29].

In this way, this protocol aims to ensure the methodology and reliability of the systematic review of self-management instruments for individuals with T2DM.

Although a systematic review of self-management instruments for individuals with diabetes already exists [13], it is limited to specific diabetes instruments. Moreover, it does not include instruments developed and validated for chronic diseases (including T2DM) and is restricted to English and Mandarin-language instruments. The proposed review protocol has significant strengths, including instruments in multiple languages and a content analysis of instrument items linked to the ICF. Such linkage is essential, as the ICF provides a standardized global language for functionality and rehabilitation.

Although the protocol is methodologically adequate for the construction of a systematic review, some limitations may be found. One of them is the heterogeneity of self-management instruments, since we have a vast number of specific and generic tools for T2DM. To minimize this limitation, the instruments will be grouped and categorized into 10 groups. This categorization will help the authors in the review construction and the readers to identify instruments according to their relevant domains.

Besides the limitations, the authors believe the review will be able to provide a comprehensive overview of the instruments available for this population in different contexts, their measurement properties and content regarding the ICF. Consequently, we believe that the use of appropriate instruments may improve clinical and functional assessment which contributes to better treatment choices and rehabilitation programs.

Upon completing the review, we aim to provide a summarized and critically evaluated compilation of self-management instruments for individuals with T2DM. This document will guide researchers and healthcare professionals in selecting the most suitable instrument based on its measurement properties and relevance to their target population.

## Conclusion

In conclusion, this review will fill in an important gap in the literature by synthesizing all self-management tools for individuals with T2DM. The expected results will contribute not only to future research but will also have clinical implications, such as better assessment and interventions planning in the care of individuals with T2DM.

## Supporting information

**S1 File. Search Strategies.**
(DOCX)

**S2 File. Table_PRISMA-P 2015 Checklist.**
(DOCX)

## Author contributions

**Conceptualization:** José Alexandre Barbosa de Almeida, Ana Tereza do Nascimento Sales Figueiredo Fernandes, Lucien Peroni Gualdi.

**Data curation:** José Alexandre Barbosa de Almeida, Thayla Amorim Santino, Rêncio Bento Florêncio.

**Formal analysis:** José Alexandre Barbosa de Almeida, Karolinne Souza Monteiro, Rêncio Bento Florêncio.

**Investigation:** José Alexandre Barbosa de Almeida, Ana Tereza do Nascimento Sales Figueiredo Fernandes, Lucien Peroni Gualdi.

**Methodology:** José Alexandre Barbosa de Almeida, Karolinne Souza Monteiro, Thayla Amorim Santino.

**Supervision:** Ana Tereza do Nascimento Sales Figueiredo Fernandes, Lucien Peroni Gualdi.

**Visualization:** José Alexandre Barbosa de Almeida, Karolinne Souza Monteiro, Thayla Amorim Santino, Rêncio Bento Florêncio, Ana Tereza do Nascimento Sales Figueiredo Fernandes, Lucien Peroni Gualdi.

**Writing – original draft:** José Alexandre Barbosa de Almeida.

**Writing – review & editing:** Ana Tereza do Nascimento Sales Figueiredo Fernandes, Lucien Peroni Gualdi.

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
