## [Decision Letter · Decision Letter 0]

28 May 2025

Dear Dr. Almeida,

Thank you for submitting your manuscript to PLOS ONE. After careful consideration, we feel that it has merit but does not fully meet PLOS ONE’s publication criteria as it currently stands. Therefore, we invite you to submit a revised version of the manuscript that addresses the points raised during the review process.

We look forward to receiving your revised manuscript.

Kind regards,

Manuela Mendonça Figueirêdo Coelho, Ph.D

Academic Editor

PLOS ONE

Journal Requirements:

2. Thank you for stating in your Funding Statement: [This study was partly financed by the Coordenação de Aperfeiçoamento de Pessoal de Nível Superior – Brasil (CAPES). Fiance Code 001.].

Reviewers' comments:

Reviewer's Responses to Questions

**Comments to the Author**

1. Does the manuscript provide a valid rationale for the proposed study, with clearly identified and justified research questions?

Reviewer #1: Yes

Reviewer #2: Partly

Reviewer #3: Yes

2. Is the protocol technically sound and planned in a manner that will lead to a meaningful outcome and allow testing the stated hypotheses?

Reviewer #1: Yes

Reviewer #2: Partly

Reviewer #3: Yes

3. Is the methodology feasible and described in sufficient detail to allow the work to be replicable?

Reviewer #1: Yes

Reviewer #2: Yes

Reviewer #3: Yes

4. Have the authors described where all data underlying the findings will be made available when the study is complete?

Reviewer #1: Yes

Reviewer #2: Yes

Reviewer #3: No

5. Is the manuscript presented in an intelligible fashion and written in standard English?

Reviewer #1: Yes

Reviewer #2: Yes

Reviewer #3: Yes

You may also provide optional suggestions and comments to authors that they might find helpful in planning their study.

Reviewer #1: Thank you for the opportunity to contribute as a reviewer for this manuscript. I believe this review is important and valuable for advancing the field of T2DM care.

Abstract:

• It is a good, concise introduction. I think it would be better to be more clearly specifying the importance of self-management in relation to T2DM. The current version makes me understand that self-management is essential for disease control in general e.g. other chronic diseases (not specific to T2DM).

• The last sentence mentioning the use of ICF as a guide for data analysis which I think it might better fit under the “Methods” rather than “Context and Objectives”.

Introduction:

• The introduction is concise and easy to understand. However, in line 78-84, could you explain more the rationale for the need of this study according to the limitation of previous study (limitations regarding the inclusion of instruments for individuals with T2DM) and show how your review addresses this gap in knowledge.

Methods:

• Eligibility criteria: I am wondering why study designs such as cross-sectional studies and post-intervention analyses will be excluded, and what study design or research type will be included in this review (e.g. psychometric study)? I understand that this review is focussing on the study targeting the development and validation of instrument. In case of the paper regarding instrument development were reported in qualitative study, would it be included or excluded?

• Search strategy:

o Would the hand search of the reference lists (backward citation tracking) of all study mean those study met eligibility criteria, or all study retrieved from databases?

o I am thinking that forward citation tracking could also be beneficial for this review.

o Well-structured and thorough search terms (according to Supplementary File 1).

Certainty of evidence and instrument recommendation: The instrument recommendation within this review is very important and it will benefit research and practice in the field of T2DM.

Data synthesis:

• You mentioned that you will use a narrative synthesis. My thinking is that it would be good to give citation to make it clearer that which (version of) narrative synthesis you will be using. I also think it could be helpful for reader if you can provide a very short list/steps of narrative synthesis you are planning to do.

• It is a good point that you mentioned “If a validation study of the same instrument for different populations exists, its measurement properties results will be considered as a single instrument”. I noticed that one of your inclusion criteria is to include cultural adaptation study and that I would assume that you might include this kind of study which may have different language versions of instrument. It would be good to mention this as well.

Discussion: Clear justification to conduct this review and provided implication of the result of this review.

Reviewer #2: The manuscript is well-organized for a systematic review. The design follows recognized guidelines (COSMIN, PRISMA-P, and GRADE), reflecting good methodological rigor. However, there are areas that need improvement.

Abstract:

The objective is valid but could be rephrased to be clearer and more concrete. It would be ideal to add a final sentence summarizing what is expected to learn or contribute with this review. Additionally, including main results and conclusions is recommended.

Introduction:

The study’s justification is relevant, but it could be enriched with a brief overview of existing instruments for self-management in T2DM, as well as a critical reflection on previous reviews. This would help better position the manuscript’s added value.

Methods:

Search strategy: It would be useful to specify an estimated cutoff date for the bibliographic searches.

Contact with authors: Though mentioned later, it would be worth clarifying whether any contact was made with the authors of the reviewed articles and under what terms.

Results:

As a systematic review, it would be interesting to include the results of each search conducted across different databases, along with a brief outline of how the instruments will be classified.

Discussion:

The current section is somewhat generic. To strengthen it, we could delve deeper into the clinical and methodological implications of the study, anticipate potential limitations, and highlight how this review surpasses or complements previous work. Also, consider adding conclusions from the systematic review.

Finally, the basis for classifying the reviewed instruments and their analysis is unclear.

Reviewer #3: General assessment

This is a well-designed and timely protocol for a systematic review of self-management instruments in type 2 diabetes mellitus (T2DM), including an evaluation of their psychometric properties and linkage to the International Classification of Functioning, Disability and Health (ICF). The protocol is clearly structured, adheres to COSMIN guidelines, and proposes a comprehensive search strategy.

The inclusion of instrument databases such as PROMIS and PROQOLID is a notable strength, and the authors’ decision to assess content via ICF coding adds an innovative dimension. The protocol addresses an important topic with practical implications for clinical and research contexts, particularly in the identification of reliable and valid tools for assessing self-management in chronic illness.

Major points for clarification and improvement

1. Instruments with subscales for self-management

The exclusion of instruments measuring more than one construct is understandable; however, many validated instruments include distinct subscales for self-management. It is unclear whether such tools will be excluded altogether or considered based on the relevance of the subscale. The authors are encouraged to clarify this point, as it could impact the comprehensiveness of the review.

2. Choice of databases – rationale for excluding Cochrane and PEDro

The protocol omits both the Cochrane Library and PEDro, despite their relevance. Cochrane CENTRAL could include validation studies nested within trials, and PEDro may contain relevant psychometric evaluations in physiotherapy contexts. If these databases were deliberately excluded, the authors should briefly justify this decision.

3. Lack of explicit quality thresholds for instrument evaluation

While the authors refer to COSMIN and GRADE frameworks, there is no indication of the thresholds that will be used to assess whether an instrument demonstrates adequate reliability, validity, or responsiveness (e.g., Cronbach’s alpha ≥ 0.70). Including these definitions would strengthen the transparency and reproducibility of the review.

4. Handling of shortened or alternate versions of instruments

The protocol does not specify how the authors will deal with instruments that have short forms or multiple validated versions. Will these be reviewed separately, combined, or will only the original versions be considered? This decision should be made explicit.

5. Purpose and operationalization of ICF linkage

Although the plan to link instrument content to the ICF is an interesting innovation, the rationale and utility of this approach remain vague. Will this linkage be used to recommend instruments for specific clinical needs or populations? How will it inform the interpretation or selection of tools?

6. Language strategy in search procedures

The authors state that there will be no language restrictions, but the search strategies do not indicate how non-English terms will be incorporated. Will search terms be translated into other languages? Will translators be used to assess studies in different languages? Clarifying the technical implementation of this inclusive strategy would be helpful.

7. Lack of data sharing plan

The protocol does not specify whether and where the extracted data, quality assessments, or ICF mappings will be made available upon completion of the review. Including a plan for data sharing (e.g. via OSF, Figshare, or journal supplements) would align the protocol with open science practices and enhance its utility for researchers and practitioners.

Summary

This is a promising and well-constructed protocol with strong methodological foundations. Addressing the six points listed above would enhance its clarity, transparency, and completeness, ensuring the resulting review is as rigorous and comprehensive as possible.

**Do you want your identity to be public for this peer review?** For information about this choice, including consent withdrawal, please see our Privacy Policy

Reviewer #1: **Yes: ** Natthapon Inta

Reviewer #2: No

Reviewer #3: **Yes: ** Andrzej Śliwerski

---

## [Author Response · Author response to Decision Letter 1]

14 Jul 2025

PONE-D-25-17107

Manuscript entitled: “Assessment of the Psychometric Properties of Self-Management Measurement Instruments for Individuals with Type 2 Diabetes: A Systematic Review Protocol”.

Dear editor and reviewers,

Thank you very much for your suggestion on our manuscript. We carefully considered your comments and revised the paper based on those comments and recommendations. We hope that these revisions improve the quality of the paper. You will find detailed responses to the comments made by you as well as those of the reviewers below.

Editor(s)' Comments

Authors’ response: we appreciate your suggestion and ensure that we have checked all style requirements to meet the journal rules.

2. Thank you for stating in your Funding Statement: [This study was partly financed by the Coordenação de Aperfeiçoamento de Pessoal de Nível Superior – Brasil (CAPES). Fiance Code 001.]. Please provide an amended statement that declares *all* the funding or sources of support (whether external or internal to your organization) received during this study, as detailed online in our guide for authors at http://journals.plos.org/plosone/s/submit-now. Please also include the statement “There was no additional external funding received for this study.” in your updated Funding Statement. Please include your amended Funding Statement within your cover letter. We will change the online submission form on your behalf.

Authors’ response: We appreciate your concern. When we declare that “This study was partly financed by the Coordenação de Aperfeiçoamento de Pessoal de Nível Superior – Brasil (CAPES). Finance Code 001” we mean that Alexandre Almeida, as a post-graduation student, receives a scholarship that covers his PhD fees. There is no additional funding for this study as well as CAPES has no interference with the publications. It is worth highlighting that the statement is in accordance with the institution rules for publication. In this way we have included the following statement in the revised cover letter.

“This study was partly financed by the Coordenação de Aperfeiçoamento de Pessoal de Nível Superior – Brasil (CAPES). Finance Code 001. This funding source had no role in the preparation or decision of the manuscript . There was no additional external funding received for this study.”

Authors’ response: we appreciate your concern and agree that if we had any data to share at this moment it would be worth having an established plan to make them public. However, as a systematic review protocol we have no data available yet as the search did not start. As soon as we start data collection we will make a data share plan. In this way we would like to apologize if we have made any mistake by filling in this submission section. Moreover, we kindly ask the editor to consider our explanation and disregard the information filled during the submission process.

Reviewer #1

Thank you for the opportunity to contribute as a reviewer for this manuscript. I believe this review is important and valuable for advancing the field of T2DM care.

Abstract:

Comment 1: It is a good, concise introduction. I think it would be better to be more clearly specifying the importance of self-management in relation to T2DM. The current version makes me understand that self-management is essential for disease control in general e.g. other chronic diseases (not specific to T2DM).

Authors’ response: We appreciate your suggestion and have added some other information regarding self-management for the T2DM population specifically. Such inclusion may be seen on page 02, lines 34-36.

Comment 2: The last sentence mentioning the use of ICF as a guide for data analysis which I think it might better fit under the “Methods” rather than “Context and Objectives”.

Authors’ response: We appreciate your concern with this information. However, one of the secondary aims of the study is to analyze the extracted content of the items and link them to the International Classification of Functioning, Disability, and Health (ICF). In this way, it is mentioned at the end of the introduction session where we describe the study aims (page 02, lines 36-39). Moreover, it is also mentioned in the methods session ( page 02, lines 51-54).

Introduction:

Comment 3: The introduction is concise and easy to understand. However, in line 78-84, could you explain more the rationale for the need of this study according to the limitation of previous study (limitations regarding the inclusion of instruments for individuals with T2DM) and show how your review addresses this gap in knowledge.

Authors’ response: We appreciate your suggestion. To make it clear we have better explained the limitations of the previous studies. Moreover, the methodology shows how we plan to fill these gaps as we are including generic instruments that included T2DM participants and have no language restrictions. Changes may be seen on page 03, lines 79-86.

Methods:

Comment 4: (Eligibility criteria) I am wondering why study designs such as cross-sectional studies and post-intervention analyses will be excluded, and what study design or research type will be included in this review (e.g. psychometric study)? I understand that this review is focussing on the study targeting the development and validation of instrument. In case of the paper regarding instrument development were reported in qualitative study, would it be included or excluded?

Authors’ response: We appreciate your concern with this information. However, the eligibility, inclusion and exclusion criteria were constructed following the COSMIN guidelines, which recommends the exclusion of studies that use instruments to compare outcomes (such as cohort studies, clinical trials, cross-sectional studies, post-intervention analyses, and case-control studies). Moreover, according to COSMIN* “Including them would require a more extensive and time-consuming search to find all studies on the mediation properties of any instrument in the population of interest, in addition to being unfeasible to read all studies, since some instruments used may not be reported in their abstracts.”

*Mokkink LB, Elsman EBM, Terwee CB. COSMIN guideline for systematic reviews of patient-reported outcome measures version 2.0. Qual Life Res. 2024 Nov;33(11):2929-2939. doi: 10.1007/s11136-024-03761-6. Epub 2024 Aug 28. PMID: 39198348; PMCID: PMC11541334.

Comment 5: (Search strategy) Would the hand search of the reference lists (backward citation tracking) of all study mean those study met eligibility criteria, or all study retrieved from databases? I am thinking that forward citation tracking could also be beneficial for this review. Well-structured and thorough search terms (according to Supplementary File 1).

Authors’ response: We appreciate your question. Hand search will be performed in reference lists of all primary studies, systematic reviews, and gray literature to include potential studies. Such information is shown in the search strategy session.

Comment 6: (Certainty of evidence and instrument recommendation) The instrument recommendation within this review is very important and it will benefit research and practice in the field of T2DM.

Authors’ response: We appreciate your comment and we also believe that we may bring some benefit to the practice in T2DM field.

Comment 7: (Data synthesis) You mentioned that you will use a narrative synthesis. My thinking is that it would be good to give citation to make it clearer that which (version of) narrative synthesis you will be using. I also think it could be helpful for reader if you can provide a very short list/steps of narrative synthesis you are planning to do.

Authors’ response: We appreciate your concern and agree that a list of steps may help the reader to understand such information. In this way, we have added on page 6, lines 190 to 195 of the revised manuscript the following paragraph:

“Additionally, the narrative synthesis will be conducted according to the following steps: 1) grouping studies by construct and measurement properties assessed; 2) describing the methodological quality of each study based on the COSMIN Risk of Bias checklist; 3) summarizing the results for each measurement property, including consistency across studies; and 4) drawing overall conclusions for each measurement property per instrument, in accordance to the COSMIN* and GRADE recommendations.”

Besides the narrative synthesis the measurement properties will be rated as s sufficient (+), insufficient (-), inconsistent (±), or indeterminate (?). The final rating of a measurement property will correspond to its lowest rating. The complete information is written on page 6, lines 196 to 200.

*Mokkink LB, Elsman EBM, Terwee CB. COSMIN guideline for systematic reviews of patient-reported outcome measures version 2.0. Qual Life Res. 2024 Nov;33(11):2929-2939. doi: 10.1007/s11136-024-03761-6. Epub 2024 Aug 28. PMID: 39198348; PMCID: PMC11541334.

Comment 8: It is a good point that you mentioned “If a validation study of the same instrument for different populations exists, its measurement properties results will be considered as a single instrument”. I noticed that one of your inclusion criteria is to include cultural adaptation study and that I would assume that you might include this kind of study which may have different language versions of instrument. It would be good to mention this as well.

Authors’ response: We appreciate your concern. Such information is explained in the data synthesis session as shown bellow and on page 06, lines 185-188.

"If a validation study of the same instrument for different populations exists, its measurement properties results will be considered as a single instrument, but specific characteristics of each version will be discussed"

Discussion:

Comment 9: Clear justification to conduct this review and provided implication of the result of this review.

Authors’ response: Thank you for your comment.

Reviewer #2

The manuscript is well-organized for a systematic review. The design follows recognized guidelines (COSMIN, PRISMA-P, and GRADE), reflecting good methodological rigor. However, there are areas that need improvement.

Abstract:

Comment 1: The objective is valid but could be rephrased to be clearer and more concrete. It would be ideal to add a final sentence summarizing what is expected to learn or contribute with this review. Additionally, including main results and conclusions is recommended.

Authors’ response: we appreciate your suggestion. The aim was rephrased as seen in page 02, lines 36-39 as well as a discussion topic was added to explain the contributions of the review (page 02, lines 54-56). On the other hand we were not able to add results and conclusions as this is a systematic review protocol study.

Introduction:

Comment 2: The study’s justification is relevant, but it could be enriched with a brief overview of existing instruments for self-management in T2DM, as well as a critical reflection on previous reviews. This would help better position the manuscript’s added value.

Authors’ response: Thank you for your observation. We have made the recommended adjustments, improving the justification in a reflective way on what is available in the literature and the gaps that still exist (page 03, lines 79-86).

Methods:

Comment 3: (Search strategy) It would be useful to specify an estimated cutoff date for the bibliographic searches.

Authors’ response: We appreciate your suggestion however, considering that there is no date restriction for the protocol we did not include a cutoff point for it. Moreover, the final review will bring this information on the results session.

Comment 4: (Contact with authors) Though mentioned later, it would be worth clarifying whether any contact was made with the authors of the reviewed articles and under what terms.

Authors’ response: We appreciate your suggestion. However, considering that this is the systematic review protocol we did not start the search strategy. In this way we do not know yet if we will need to contact any authors or if all documents will be available. In case we need to contact any authors we will send an email asking for the full document.

Results:

Comment 5: As a systematic review, it would be interesting to include the results of each search conducted across different databases, along with a brief outline of how the instruments will be classified.

Authors’ response: Thank you for your concern. This information cannot be included as we have not started the searches yet. Since this is only a review protocol, we are following the schedule made during its registration at PROSPERO (CRD42024605840). However, the systematic review, when published, will bring all information regarding your questioning.

Discussion:

Comment 6: The current section is somewhat generic. To strengthen it, we could delve deeper into the clinical and methodological implications of the study, anticipate potential limitations, and highlight how this review surpasses or complements previous work. Also, consider adding conclusions from the systematic review.

Authors’ response: We appreciate your suggestion. The discussion section was rewritten according to suggestions and included more detailed information regarding the benefits of the review and its limitations. Such changes may be seen on page 7, lines 232-233 and page 8, lines 234-259. Regarding the suggestion to add conclusions from the systematic review we highlight that this is the review protocol. In this way, we have not started data collection and analysis which does not allow us to make any conclusion. Moreover, this study is a systematic review protocol and similar studies already published present the topic of discussion superficially, since there is no data to be discussed and even no conclusion.

Comment 7: Finally, the basis for classifying the reviewed instruments and their analysis is unclear.

Authors’ response: We appreciate your concern. We highlight that each instrument will be classified and justified into one of three categories according to the COSMIN recommendations as well as their analysis. Based on this recommendation the categories are grouped as: (A) The instrument is recommended for use, and the results are reliable; (B) The instrument may be recommended but requires further research to assess its quality; (C) The instrument should not be recommended. We will also provide suggestions based on feasibility considerations. This information is shown on page 7, lines 219-224. The reference used to classify the categories is shown bellow:

Mokkink LB, Elsman EBM, Terwee CB. COSMIN guideline for systematic reviews of patient-reported outcome measures version 2.0. Qual Life Res. 2024 Nov;33(11):2929-2939. doi: 10.1007/s11136-024-03761-6. Epub 2024 Aug 28. PMID: 39198348; PMCID: PMC11541334.

Reviewer #3

General assessment:

Comment 1: This is a well-designed and timely protocol for a systematic review of self-management instruments in type 2 diabetes mellitus (T2DM), including an evaluation of their psychometric properties and linkage to the International Classification of Functioning, Disability and Health (ICF). The protocol is clearly structured, adheres to COSMIN guideli

---

## [Editor Report · Decision Letter 1]

1 Aug 2025

Assessment of the Psychometric Properties of Self-Management Measurement Instruments for Individuals with Type 2 Diabetes: A Systematic Review Protocol

PONE-D-25-17107R1

Dear Dr. Almeida,

We’re pleased to inform you that your manuscript has been judged scientifically suitable for publication and will be formally accepted for publication once it meets all outstanding technical requirements.

Kind regards,

Manuela Mendonça Figueirêdo Coelho, Ph.D

Academic Editor

PLOS ONE

Additional Editor Comments (optional):

Dear Authors,

We are pleased to inform you that your revised manuscript has been accepted for publication in PLOS ONE.

The revisions made in response to the editorial and peer reviewer comments have significantly improved the clarity, methodological rigor, and transparency of the protocol. We commend your team for the thoughtful and thorough responses provided, as well as for the scientific quality and relevance of the proposed systematic review.

Your study addresses a important gap in the literature by mapping and evaluating self-management instruments for individuals with type 2 diabetes mellitus, and we believe it will make a valuable contribution to both research and clinical practice.

Congratulations on the acceptance of your manuscript. You will receive a follow-up message from the production team regarding the final steps for publication.

Thank you for choosing PLOS ONE as the venue for your work.
---

## [Editor Report · Acceptance letter]

PONE-D-25-17107R1

PLOS ONE

Dear Dr. Almeida,

I'm pleased to inform you that your manuscript has been deemed suitable for publication in PLOS ONE. Congratulations! Your manuscript is now being handed over to our production team.

Kind regards,

on behalf of

Dr. Manuela Mendonça Figueirêdo Coelho

Academic Editor

PLOS ONE